# How Ceramides Orchestrate Cardiometabolic Health—An Ode to Physically Active Living

**DOI:** 10.3390/metabo11100675

**Published:** 2021-09-30

**Authors:** Justin Carrard, Hector Gallart-Ayala, Nadia Weber, Flora Colledge, Lukas Streese, Henner Hanssen, Christian Schmied, Julijana Ivanisevic, Arno Schmidt-Trucksäss

**Affiliations:** 1Division of Sports and Exercise Medicine, Department of Sport, Exercise and Health, University of Basel, Birsstrasse 320B, 4052 Basel, Switzerland; lukas.streese@unibas.ch (L.S.); henner.hanssen@unibas.ch (H.H.); arno.schmidt-trucksaess@unibas.ch (A.S.-T.); 2Metabolomics Platform, Faculty of Biology and Medicine, University of Lausanne, Quartier UNIL-CHUV, Rue du Bugnon 19, 1005 Lausanne, Switzerland; hector.gallartayala@unil.ch (H.G.-A.); julijana.ivanisevic@unil.ch (J.I.); 3Medical School, Department of Health Sciences and Technology, Swiss Federal Institute of Technology, Universitätstrasse 2, 8092 Zurich, Switzerland; naweber@student.ethz.ch; 4Division of Sports Science, Department of Sport, Exercise and Health, University of Basel, Birsstrasse 320B, 4052 Basel, Switzerland; flora.colledge@unibas.ch; 5Sports Cardiology Section, Department of Cardiology, University Heart Center Zurich, University Hospital Zurich, University of Zurich, 8091 Zurich, Switzerland; christian.schmied@usz.ch

**Keywords:** ceramides, sphingolipids, cardiometabolic health, cardiovascular health, metabolism, insulin, physical activity, exercise, cardiorespiratory fitness

## Abstract

Cardiometabolic diseases (CMD) represent a growing socioeconomic burden and concern for healthcare systems worldwide. Improving patients’ metabolic phenotyping in clinical practice will enable clinicians to better tailor prevention and treatment strategy to individual needs. Recently, elevated levels of specific lipid species, known as ceramides, were shown to predict cardiometabolic outcomes beyond traditional biomarkers such as cholesterol. Preliminary data showed that physical activity, a potent, low-cost, and patient-empowering means to reduce CMD-related burden, influences ceramide levels. While a single bout of physical exercise increases circulating and muscular ceramide levels, regular exercise reduces ceramide content. Additionally, several ceramide species have been reported to be negatively associated with cardiorespiratory fitness, which is a potent health marker reflecting training level. Thus, regular exercise could optimize cardiometabolic health, partly by reversing altered ceramide profiles. This short review provides an overview of ceramide metabolism and its role in cardiometabolic health and diseases, before presenting the effects of exercise on ceramides in humans.

## 1. Introduction

According to the Global Burden of Diseases, Injuries and Risk Factor Study, the rate of mortality from noncommunicable diseases (NCD) has increased from 1990 to 2019, to account for 74% of total mortality [1,2]. Strikingly, about one-half of NCD-related deaths are still attributable to cardiometabolic diseases (CMD), encompassing cardiovascular diseases (CVD), type 2 diabetes mellitus (T2DM) and non-alcoholic fatty liver disease (NAFLD) [3]. Whereas age-standardized death rates for CMD have decreased since 1990 in high-income countries, absolute number of CMD-related deaths are still increasing in these regions, mainly due to population aging [4]. Simultaneously, about two thirds of all CMD-related deaths are now occurring in low- and middle-income countries [5]. To combat the growing burden of CMD worldwide, it is urgent to improve patients’ stratification with respect to cardiometabolic health decline and CMD onset, which will enable the development of more precise and personalized prevention and treatment strategies [5].

Blood lipid analysis has been part of cardiometabolic risk stratification strategy since the middle of the last century [6]. Despite technological advances in mass-spectrometry and bioinformatics, clinical lipid analysis is still mostly restricted to total triglycerides, cholesterol, low-density lipoprotein cholesterol (LDL-C) and high-density lipoprotein cholesterol (HDL-C) [7,8]. Although these biomarkers provide an acceptable risk assessment, growing evidence shows that lower-abundant specific sphingolipid species, known as ceramides, predict cardiometabolic outcomes more precisely in patients with and without coronary artery disease [9,10,11,12]. As ceramides could eventually replace cholesterol in clinical practice [13], it is important to determine whether and how physical activity, as a powerful prevention and treatment strategy, can influence ceramide levels.

Regular physical activity is, indeed, essential to maintain general health [14,15,16,17], prevent and treat CVD [18,19,20,21], insulin resistance and T2DM [22,23,24] as well as NAFLD [25]. Adding to its appeal, physical activity is a simple, low-cost and patient-empowering means to reduce CMD-related burden in low-income, middle-income, and high-income countries [26]. Exercise, as a subset of physical activity that is structured in order to improve or maintain physical fitness [27], not only mitigates traditional risk factors for CVD but also directly improves cardiometabolic health. For instance, exercise optimizes vascular endothelial function, stimulates secretion of cardioprotective myokines, but also acts through other mechanisms not fully elucidated yet, which might include changes in ceramide metabolism [19]. This short review provides an overview on ceramide metabolism and its role in human cardiometabolic health and diseases before presenting the effects of physical exercise on ceramides.

## 2. Ceramide Metabolism

Ceramides are synthesized through three major metabolic pathways: the de novo pathway that mainly results from the combination of serine and palmitoyl-CoA in the endoplasmic reticulum, the salvage pathway, which consists of the breakdown of sphingomyelins or glycosphingolipids in lysosomes, and the sphingomyelinase pathway, where sphingomyelins are converted into ceramides in cell membrane (Figure 1) [28,29]. Due to the specific subcellular localization of the enzymes controlling ceramides metabolism, the latter is highly compartmentalized [29].

In the Golgi apparatus, ceramides are transformed into various complex sphingolipids such as ceramide-1-phosphate, sphingomyelin or glycosphingolipids [28,29] Ceramides can also be acylated to form acylceramides, which accumulate in intracellular lipid droplets [29,30]. Finally, the only ‘exit’ from ceramide metabolism results in the production of fatty aldehydes and ethanolamine phosphate by the action of a sphingosine-1-phosphate (S1P) lyase on sphingoid base phosphates [29].

Less frequently, substrates other than serine and palmitoyl-CoA, such as alanine or glycine plus stearate or myristate, can also enter the de novo pathway [29]. This leads to the formation of structurally different sphingoid bases, namely deoxysphingoid and 1-deoxymethyl sphingoid bases [29]. Distinct isoforms of ceramide synthases also add variations in the acyl groups of sphingoid bases, which further increases diversity and complexity of ceramides structures [29]. Additionally, hydroxylases and desaturases can modify acyl groups; thus, ceramides should be regarded as a family of related but structurally distinct lipid species [29]. These structural variations are at the origin of the distinct biological functions attributed to the different ceramide species [29].

## 3. Ceramides in Cardiometabolic Health and Diseases

Ceramides and their derivatives are involved in many, if not all, essential cellular processes such as cell growth, cell adhesion and migration, senescence, apoptosis, inflammation, immune responses, and angiogenesis [28,29]. They are also part of cell membrane, where they influence vesicular transport, membrane receptors, and exosome secretion [29]. Thus, ceramides are implicated in both intracellular and cell-to-cell signaling [31]. Ceramides synthesis and accumulation are influenced by multiple factors, which include excessive supply of substrates, systemic inflammation, oxidative stress and the microbiome [31]. In obesity, levels of Tumor Necrosis Factor-Alpha (TNF-α) are elevated, and Toll-like receptors (TLRs) are stimulated by excess of fatty acids [32]. Both TNF-α and TLRs can stimulate ceramide synthesis enzymes [32]. Ceramides will then activate inflammasomes (specifically the NLRP3 inflammasome in adipocytes), which lead to an increase in pro-inflammatory cytokine secretion [32]. This inflammatory response can be modulated by the adipokine adiponectin through activation of the ceramidase activity of the adiponectin receptors [33]. Therefore, the many roles played by ceramides in cardiometabolic health and diseases are associated with this connection between excessive supply of lipids and inflammation [32].

Changes in ceramide levels have been observed in diverse pathological processes such as NAFLD [34,35], obesity [36,37], insulin resistance [38,39], vascular inflammation, and atherosclerosis (Figure 2) [40,41,42]. Patients suffering from NAFLD display elevated levels of circulating and liver ceramides [34]. Hepatocellular ceramides stimulate fatty acid uptake, by increasing the translocation of fatty acid transporters to the plasma membrane, and facilitate their transformation into intracellular triglycerides [34]. Ceramides also decrease glucose uptake, by inhibiting Akt/protein kinase B (PKB) activity, which inhibits translocation of glucose transporters (GLUT4) to the cell membrane and reduce hepatic gluconeogenesis [34,39,43]. This enables hepatocytes to favor fatty acids over glucose as a preferred energy source and leads to NAFLD [34].

Elevated circulating ceramide levels have been reported in obese patients with T2DM [44,45]. Furthermore, LDL were shown to be enriched in ceramides in lean patients with T2DM compared to healthy individuals [46]. In fact, an accumulation of ceramides in muscle cells and adipocytes also inhibits Akt/PKB, reduces GLUT4 translocation to the membrane, and ultimately decreases glucose uptake [43,47,48]. In adipocytes, ceramides also slow down fatty acid oxidation by inhibiting specific hormone-sensitive lipases [49]. This results in peripheral insulin resistance [42]. In addition, ceramide accumulation is believed to induce β-cell dysfunction and reduce insulin secretion [50,51]. Looking at species level, long-chain ceramides, such as Cer (18:1;2/16:0) or Cer (18:1;2/18:0), have been shown to correlate the most with insulin resistance and NAFLD [52,53]. Treatment with the glucagon-like peptide-1 receptor agonist liraglutide was recently shown to reduce ceramide levels in patient with T2DM [54]. The authors of this study then hypothesized that the cardiovascular benefit of this treatment observed within patients with T2DM might be partially due to this reduction in ceramide content [54].

From a cardiovascular perspective, ceramides located at the surface of LDL drive their transcytosis through the endothelium and uptake into macrophages, which results in foam cell formation, vascular inflammation, and atherosclerosis [40,41,42]. Thoracic adipose tissue of obese individuals secretes a particular ceramide species, named Cer (18:1;2/16:0), which acts on endothelial cells to reduce vasodilatation, induce inflammation, oxidative stress, and finally increase risk of cardiovascular death [55]. Again, circulating long-chain ceramides, such as Cer (18:1;2/16:0), Cer (18:1;2/18:0) and Cer (18:1;2/24:1), seem to be the species most frequently associated with cardiovascular morbidity and mortality [10,31,56]. Statin therapy showed conflicting results on ceramide levels. A recent study reported no reduction in absolute ceramide content within LDL particles, or even an increase of its relative concentration as other lipid species composing LDL, such as lysophospholipids, were reduced [57]. The authors of this study hypothesized that ceramides might be responsible for residual cardiovascular risk on statin therapy [57]. Conversely, other studies found a decrease in circulating ceramide levels on simvastatin or rosuvastatin therapy [58,59] as well as on proprotein convertase subtilisin/kexin type 9 (PCSK9) inhibitors [60]. Ezetimibe therapy did not lead to a reduction in circulating ceramides [59]. Thus, larger population trials are necessary to unravel effects of lipid-lowering drugs on ceramide levels [61].

## 4. Ceramide Measurement in Clinical Research and Practice

Liquid chromatography-tandem mass spectrometry (LC-MS/MS) is the technique of choice for ceramide measurement as it offers a high sensitivity to detect low abundant species and a large breadth of coverage with respect to chemical diversity [62,63]. Currently, hurdles remain in the translation of lipidomic output, obtained by LC-MS/MS, to clinical practice [64]. First, harmonization of data quality assessment is necessary to allow for measurements comparison between laboratories using different technologies and methodologies [64]. Several ring trials are currently ongoing to optimize and standardize this analytical aspect [64]. The next step forward is the establishment of biological reference intervals for the species, which are amenable to be used in clinical practice. Establishing these reference values through large-scale population studies, while taking variables such as age, sex, physical activity, diet, medication, and circadian rhythm into consideration, is required to distinguish physiological from pathological variations [64,65].

Nevertheless, plasma ceramides are already measured in daily practice in some clinical facilities, such as at the Mayo Clinic (Rochester, Minnesota, USA) [66,67]. Concretely, the Coronary Event Risk Test 1 (CERT1) is calculated based on concentrations of Cer (18:1;2/16:0), Cer (18:1;2/18:0), Cer (18:1/24:1) and on the ratios Cer (18:1;2/16:0) to Cer (18:1;2/24:0), Cer (18:1;2/18:0) to Cer (18:1;2/24:0) and Cer (18:1;2/24:1) to Cer (18:1;2/24:0) [9,10,11,12,56,68,69,70,71,72,73]. One or two points are attributed for each result above the median or the third quartile, respectively [66]. This results in a 12-point scale categorizing the risk of myocardial infarction, acute coronary syndromes, and mortality within 1 to 5 years [66]. This score was improved by adding some specific phosphatidylcholine (PC) species to it, resulting in the Coronary Event Risk Test 2 (CERT2) [12]. This score consists of the ratios Cer (18:1;2/24:1) to Cer (18:1;2/24:0), Cer (18:1;2/16:0) to PC (16:0/22:5), Cer (18:1;2/16:0) to PC (14:0/22:6) and the concentration of PC 16:0/16:0 [12]. CERT2 was shown to efficiently predict residual cardiovascular events in patients with coronary artery disease (CAD) [12], to identify high-risk patients among patients with stable CAD [74] and to predict the risk of cardiovascular death in patients with acute coronary syndrome, independently of established risk factors [75]. Both CERT1 and CERT2 were shown to be superior to the Systematic COronary Risk Evaluation (SCORE) with regard to prediction of cardiovascular events, cardiovascular mortality, and overall mortality in patients with CVD, with CERT2 displaying better results than CERT1 [76]. Predictive performance of CERT2 was further improved once combined with SCORE [76]. As SCORE was recently updated to SCORE2, it might be worthful to repeat these comparisons and likely to integrate ceramide and phosphatidylcholine items in the next version of SCORE [77].

Overall, circulating ceramides can predict cardiovascular outcomes beyond traditional cardiovascular risk factors in healthy individuals and patients with CAD (Figure 3) [9,10,12,78,79,80,81]. On a mechanistic level, however, ceramide metabolism, along with its role in cardiometabolic health, is not as deeply understood as that of cholesterol [13]. Therefore, further mechanistic, epidemiological, and interventional studies are required to advance this field of research [13]. Conversely to CVD, no ceramides-based risk assessment tool is available for insulin resistance, type 2 diabetes and NAFLD yet, although circulating ceramides have been associated with these pathologies in clinical cohort studies [34,44,82,83]. Finally, ceramide metabolism might represent a future therapeutic target, as exemplified by studies in rodents demonstrating that inhibition of ceramide synthesis reduces incidence of hypertension, type 2 diabetes mellitus, NAFLD, atherosclerosis and heart failure [49,61].

## 5. Exercise-A Modulator of Ceramide Levels

Physical activity and exercise represent cost-effective means to prevent and treat NCD [15,26,85,86,87]. While physical activity notoriously mitigates traditional cardiometabolic risk factors, mechanisms through which exercise directly improves cardiometabolic health remain poorly understood [19]. Preliminary data suggests that exercise could influence ceramide metabolism and thereby optimize human health.

Circulating ceramide levels were increased following a single session of moderate-intensity continuous training (MICT) in endurance athletes, sedentary obese individuals and patients with T2DM [88]. Likewise, muscle ceramide levels increased after three hours of cycling [89]. Conversely, a several-week period of MICT lowered circulating ceramide levels in patients suffering from obesity or T2DM [90,91]. Regular MICT or high-intensity interval training (HIIT) also lead to a reduction in muscle ceramides [91,92,93].

Cardiorespiratory fitness (CRF), defined as the peak oxygen uptake (VO_2_peak), is inversely associated with incidence of cancer, cardiometabolic diseases and all-cause mortality [94,95,96,97,98,99]. CRF has been shown to be a better predictor of morbidity and mortality than physical activity itself [100,101,102]. Thus, the American Heart Association even recommends assessing CRF as a vital sign in clinical practice [97]. Strikingly, several ceramide species were reported to be negatively associated with CRF: Cer (18:0;2/22:1) [103], Cer (18:0;2/24:1) [103], Cer (18:1;2/14:0) [103], Cer (18:1;2/16:0) [103,104,105], Cer (18:1;2/18:0) [105,106], Cer (18:1;2/20:0) [105,106], Cer (18:1;2/22:1) [103] and Cer (18:1;2/24:1) [105]. The three cardiometabolically deleterious ceramide species clinically used in the ceramide-phospholipid score (Cer (18:1;2/16:0), Cer (18:1;2/18:0) and Cer (18:1;2/24:1)) were notably reported to be negatively associated with CRF [12,79]. Finally, some glycosphingolipids have also been negatively associated with CRF, such as HexCer (18:1;2/16:0) [103,107], HexCer (18:1;2/18:0) [103,105], LacCer (18:1;2/18:1) [103] and LacCer (18:1;2/22:0) [103].

To summarize, acute bouts of physical activity tend to elevate both circulating and muscle ceramide levels, while regular exercise leads to a reduction of ceramides in the circulation and myocytes (Figure 4). It has been hypothesized that the increase in ceramide could inhibit insulin action, promote fatty acid oxidation and thereby favor lipids as an energy source during exercise [28]. The effect of regular exercise on ceramides is also supported by the reported negative associations between ceramides and CRF. Therefore, it can be hypothesized that regular exercise, leading to an improvement of CRF, optimizes cardiometabolic health, partly by reversing altered ceramide profiles.

## 6. Conclusions

Upgrading patients’ metabolic stratification in clinical practice has the potential to improve personalization of CMD prevention and treatment. Specific ceramide species were shown to predict cardiometabolic outcomes beyond classical biomarkers such as cholesterol. While an acute bout of physical exercise tends to increase circulating and muscular ceramide levels, regular exercise leads to a reduction of ceramide levels. This is also reflected by the fact that CRF, a powerful marker reflecting cardiometabolic health and training status, has been negatively associated with several ceramide species. Preventive and therapeutic effectiveness of a well-conducted exercise intervention to mitigate ceramides profile remains to be investigated.

## Figures and Tables

**Figure 1 metabolites-11-00675-f001:**
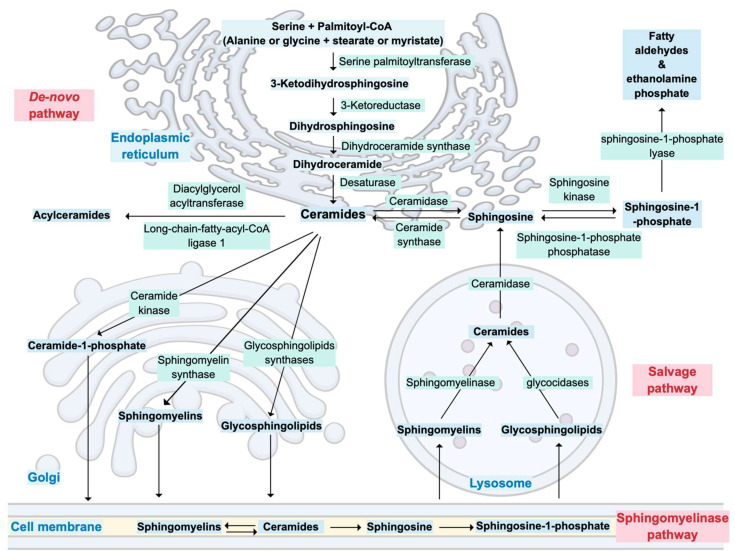
Overview of ceramide metabolism and its cellular compartmentalization.

**Figure 2 metabolites-11-00675-f002:**
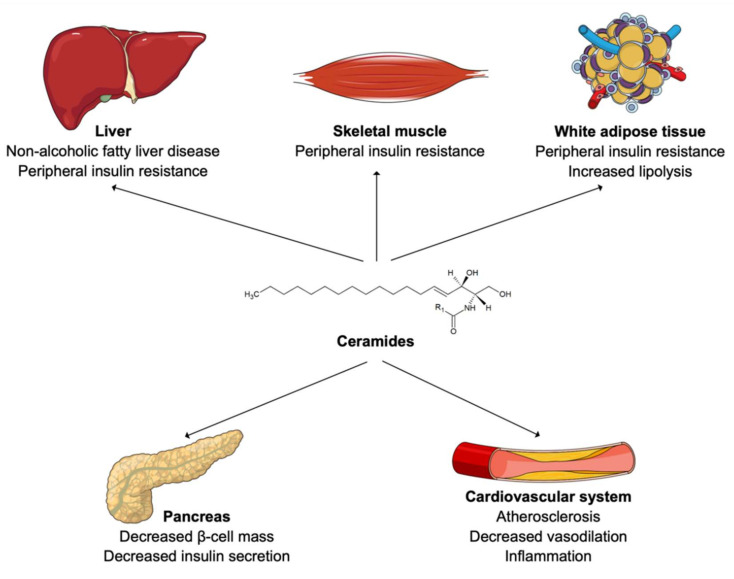
Ceramides in human cardiometabolic diseases.

**Figure 3 metabolites-11-00675-f003:**
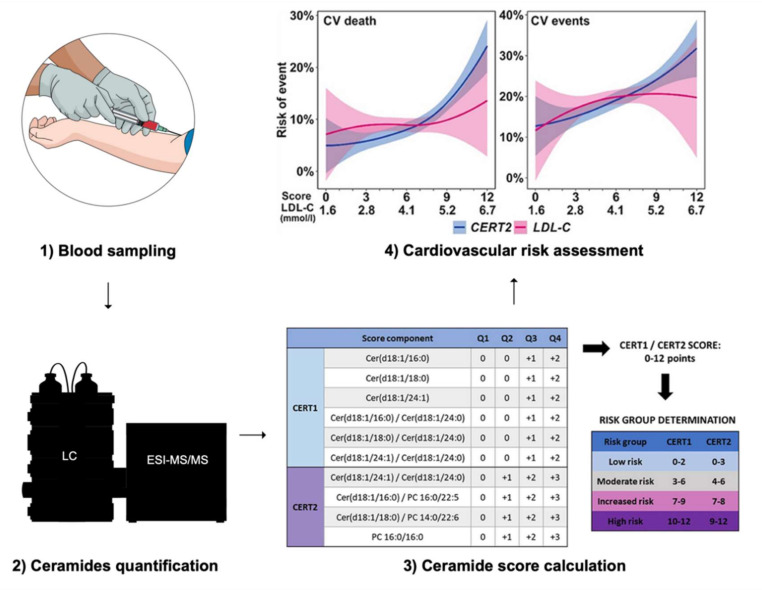
Cardiovascular risk assessment using ceramides-based scores. Abbreviations: LC = liquid chromatography, ESI-MS/MS = electrospray ionization tandem-mass spectrometry, CV = cardiovascular, LDL-C = low-density lipoprotein cholesterol, CERT1 = Ceramide Test Score 1, CERT2 = Ceramide Test Score 2. The figures illustrating the stages 3 and 4 are reproduced, without modification, under the terms of the Creative Commons Attribution License from Hilvo et al. [84].

**Figure 4 metabolites-11-00675-f004:**
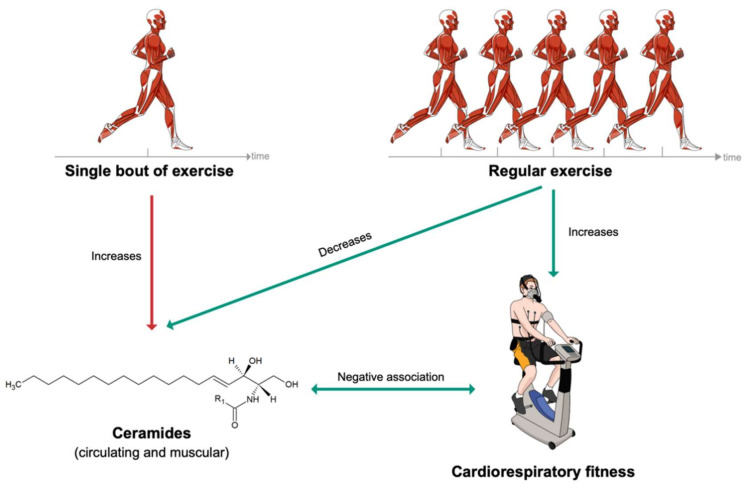
Short- and long-term effects of exercise on ceramide levels.

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
