# Peer review of "How Ceramides Orchestrate Cardiometabolic Health—An Ode to Physically Active Living"

_metabolites, 2021, doi:10.3390/metabo11100675_

Round 1

Reviewer 1 Report

I would wish to congratulate the authors on the well-executed review on this topic. It is richly equipped with images.

  1. Please change the word peripheric to the peripheral in Figure 2.
  2. Is there any evidence that some of the common cardiovascular therapies such as statins affect ceramide levels? This should be elaborated.
  3. What about the connection of ceramide metabolism with inflammatory processes? Is there a connection established?
  4. What about the effect of glucose-lowering therapies on ceramide levels, such as the one of liraglutide on ceramide levels? I would advise authors to put present pharmacotherapy in the relevant clinical context with respect to ceramide metabolism.
    Please include work such as doi: 10.1136/bmjdrc-2021-002395.
  5. Please comment on the latest work by Leiherer et al. that should be included in this work: 10.1093/eurjpc/zwab112. 

Reviewer 2 Report

This review focuses on certain types of lipids known as ceramides. A recent meta-analysis showed that higher plasma levels of Cer (d18: 1/16: 0), Cer (d18: 1/18: 0) and Cer (d18: 1/24: 1) were associated with major adverse cardiovascular events (Mantovani A, 2020). Therefore, the topic of the review is relevant and interesting for both researchers and clinicians. The review also examines the effect of physical activity and physical training on the level of ceramides, the association of the level of physical performance with the level of ceramides. In conclusion, it is proposed to evaluate the effectiveness of physical training, including the effect on the level of ceramides, which deserves further research.
